# Cloning antibodies from single cells in pooled sequence libraries by selective PCR

Felix Horns[1], Stephen R. Quake[2,3,4]*

**1** Biophysics Graduate Program, Stanford University, Stanford, California, United States of America,
**2** Department of Bioengineering, Stanford University, Stanford, California, United States of America,
**3** Department of Applied Physics, Stanford University, Stanford, California, United States of America, **4** Chan Zuckerberg Biohub, Stanford, California, United States of America

\* steve@quake-lab.org

**Data Availability Statement:** Pipeline for SPAR primer design and analysis code is freely available at https://github.com/felixhorns/spar.

**Funding:** This work was funded by the Chan-Zuckerberg Biohub. The funders had no role in

## Abstract

Antibodies function by binding to antigens. Antibodies must be cloned and expressed to determine their binding characteristics, but current methods for high-throughput antibody sequencing yield antibody DNA pooled from many cells and do not readily permit cloning of antibodies from single B cells. We present a strategy for retrieving and cloning antibody DNA from single cells within a pooled library of cells. Our strategy, called selective PCR for antibody retrieval (SPAR), takes advantage of the unique sequence barcodes attached to individual cDNA molecules during sample preparation to enable specific amplification by PCR of antibody heavy- and light-chain cDNA originating from a single cell. We show through computational analysis that most human antibodies sequenced using typical high-throughput methods can be retrieved using SPAR, and experimentally demonstrate retrieval of full-length antibody variable region cDNA from three cells within pools of ~5,000 cells. SPAR enables rapid low-cost cloning and expression of native human antibodies from pooled single-cell sequence libraries for functional characterization.

## Introduction

Antibodies enable immune recognition by binding to target molecules called antigens. Characterization of antibody binding properties, such as specificity and affinity, is essential for understanding the recognition capability of the immune system and discovering antibodies for research and therapeutics. Currently, sequence information alone is not sufficient to predict antibody specificity and affinity. Thus, characterization of antibody binding requires recombinant cloning and expression of purified protein for use in functional assays.

Single-cell approaches enable high-throughput determination of native antibody sequences, but remain inadequate for functional characterization at similar scale. Droplet- and micro-well-based single-cell sequencing techniques can identify >10,000 natively paired antibody heavy- and light-chain gene sequences per experiment [1–4]. However, current methods yield complementary DNA (cDNA) pooled from thousands of cells, rendering isolation of antibody cDNA from individual cells difficult. Based on sequence information, antibody DNA can be produced by gene synthesis [3, 5], but this approach is more costly and time-consuming than

study design, data collection and analysis, decision to publish, or preparation of the manuscript.

**Competing interests:** F.H. and S.R.Q. are inventors on a patent applied for by Stanford University and the Chan-Zuckerberg Biohub (CZB) (provisional patent application number 63/039,113) related to methods and compositions for selective PCR and cloning of antibody sequences. S.R.Q. is employed as co-president of CZB. This does not alter our adherence to PLOS ONE policies on sharing data and materials.

cDNA cloning. Single B cell sorting and reverse transcription-polymerase chain reaction (RT-PCR) directly yields antibody cDNA suitable for cloning and expression [6], but this approach lacks sufficient throughput to survey antibody sequence diversity at the scale of the immune repertoire. Thus, existing methods do not permit simultaneous high-throughput determination of antibody sequences and the rapid cloning and expression of individual antibodies, which may be chosen from the repertoire on the basis of their sequence or clonal properties, for functional characterization.

In an effort to close this methodological gap, we envisioned a strategy for cloning antibody heavy- and light-chain cDNA from a single B cell within a pooled library by leveraging the unique sequence barcodes that are attached to molecules of cDNA during sample preparation. These sequence barcodes typically include a cell barcode (CBC) used to distinguish individual cells and a unique molecular identifier (UMI) used to distinguish individual molecules of template RNA (Fig 1A). After sequencing, the antibody heavy- and light-chain sequences, and their corresponding sequence barcodes are known. We reasoned that these barcodes and the heavy- and light-chain sequences could then be used as unique molecular tags to retrieve cDNA from a single cell.

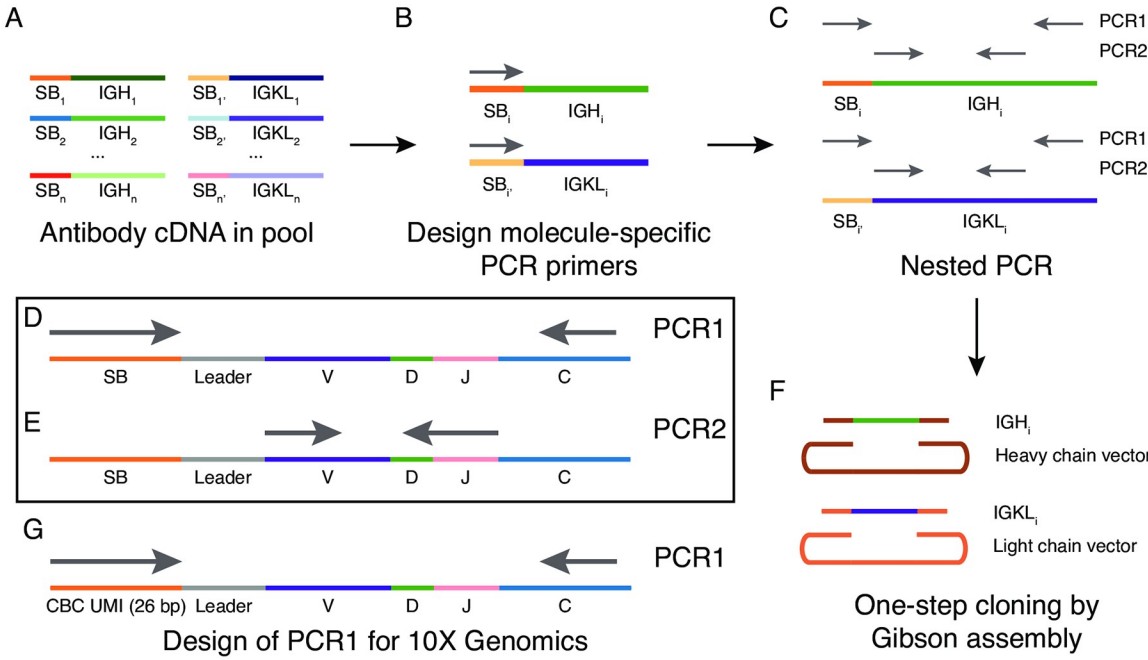

**Fig 1. Schematic of workflow for Selective PCR for Antibody Retrieval (SPAR).** (A) Antibody heavy- and light-chain cDNA (IGH and IGKL, respectively) from individual cells within a pooled library are distinguished by unique sequence barcodes (SB). For example, the heavy- and light-chain cDNA from cell 1 ($IGH_1$ and $IGKL_1$) are marked by $SB_1$ and $SB_{1'}$. (B) Molecule-specific primers are designed to target the sequence barcode. (C-E) Selective amplification of target molecules is performed by two-step nested PCR. Overview of nested PCR design for heavy (top) and light (bottom) chains in shown in (C) and details are shown in (D) and (E). In the first PCR step (PCR1), primers target the unique sequence barcode (SB) and constant region, labeled C. In the second PCR step (PCR2), primers target the 5' and 3' ends of the antibody variable region, labeled VDJ. (F) PCR products are cloned into linearized expression vectors by Gibson assembly. (G) Architecture of antibody heavy- or light-chain cDNA after library preparation using 10X Genomics Chromium 5' V(D)J platform. Sequence barcode consists of 16 bp cell barcode (CBC) and 10 bp unique molecular identifier (UMI). PCR1 forward primer targets this 26 bp sequence. PCR, polymerase chain reaction; cDNA, complementary DNA; IGH, immunoglobulin heavy chain; IGKL, immunoglobulin kappa or lambda chain; SB, sequence barcode; CBC, cell barcode; UMI, unique molecular identifier; V, variable gene; D, diversity gene; J, joining gene; C, constant gene.

## Results

### Design of Selective PCR for Antibody Retrieval (SPAR)

We designed a strategy for selective amplification of target cDNA molecules using PCR primers that specifically bind sequence barcodes (Fig 1B). Our strategy uses nested PCR consisting of two steps (Fig 1C). In the first step (PCR1), we use an outer forward primer that spans the sequence barcode, together with an outer reverse primer within the antibody constant region (Fig 1D). In the second step (PCR2), we use an inner forward primer targeting the 5' end of the variable region and an inner reverse primer targeting the 3' end of the variable region (Fig 1E). Importantly, the antibody repertoire has diversity throughout the variable region, with especially high diversity within complementarity determining region 3 (CDR3), which is located near the 3' end of the variable region, allowing PCR2 primers to enhance specificity. PCR2 yields full-length antibody variable region cDNA as a product. We designed the PCR2 primers with 5' arms that are homologous to the expression vector, enabling a simple one-step cloning procedure (Fig 1F).

We implemented this strategy for the 10X Genomics Chromium Single Cell 5' V(D)J platform. This platform uses a 16 base pair (bp) CBC and 10 bp UMI. After single-cell paired heavy- and light-chain sequencing, the complete heavy- and light-chain variable region sequences, and the corresponding CBC and UMI sequences for each cDNA molecule are known. Forward PCR1 primers were designed to target this combined 26 bp CBC and UMI sequence (Fig 1G). By performing computational primer design using Primer3 [7], we identified reverse PCR1 primers and PCR2 primer pairs that were compatible with these forward PCR1 primers and had high annealing temperature (optimally 67 C) to ensure specific amplification.

We tested this strategy using a dataset of sequences and material from our previous study [3]. This dataset consisted of 94,259 natively paired antibody heavy- and light-chain sequences obtained from single B cells, which were isolated from peripheral blood of healthy humans at 7 or 9 days after influenza vaccination. Single-cell paired heavy-light chain antibody repertoire sequencing was performed using the 10X Genomics Chromium Single Cell 5' V(D)J platform. Cells were pooled into 16 libraries, each having an average of 5,891 single cells (n = 5,891 ± 1,669, mean ± s.d., range 298–7,169 cells). For experimental validation, we used the full-length cDNA pools generated by the standard sample preparation procedure.

### SPAR primers can be designed to retrieve most of the human antibody repertoire

To assess how much of the human antibody repertoire can be retrieved using SPAR, we computationally designed SPAR primers to retrieve antibodies from all 94,259 single cells in our dataset. Successful primer design is a necessary condition for antibody retrieval. Overall, we found that SPAR primers can be designed for 81% of these cells (Fig 2A). At the single-chain level, SPAR primers can be designed for 88% of heavy-chain genes and 90% of light-chain genes (Fig 2A). Because nearly all heavy- and light-chain genes were assembled based on sequencing of multiple molecules of cDNA, most antibodies can be addressed using multiple unique barcodes (Fig 2B; median 11 UMIs per heavy chain, 21 per light chain), improving the likelihood of having at least one suitable PCR1 primer pair. These results indicate that SPAR primers can be designed to retrieve the large majority of antibodies in the human repertoire.

To assess the likely specificity of PCR1 primers targeting the sequence barcode, we examined the similarity between PCR1 forward primers for heavy-chain genes in one pooled library. We found that PCR1 forward primer sequences are substantially dissimilar (Fig 2C):

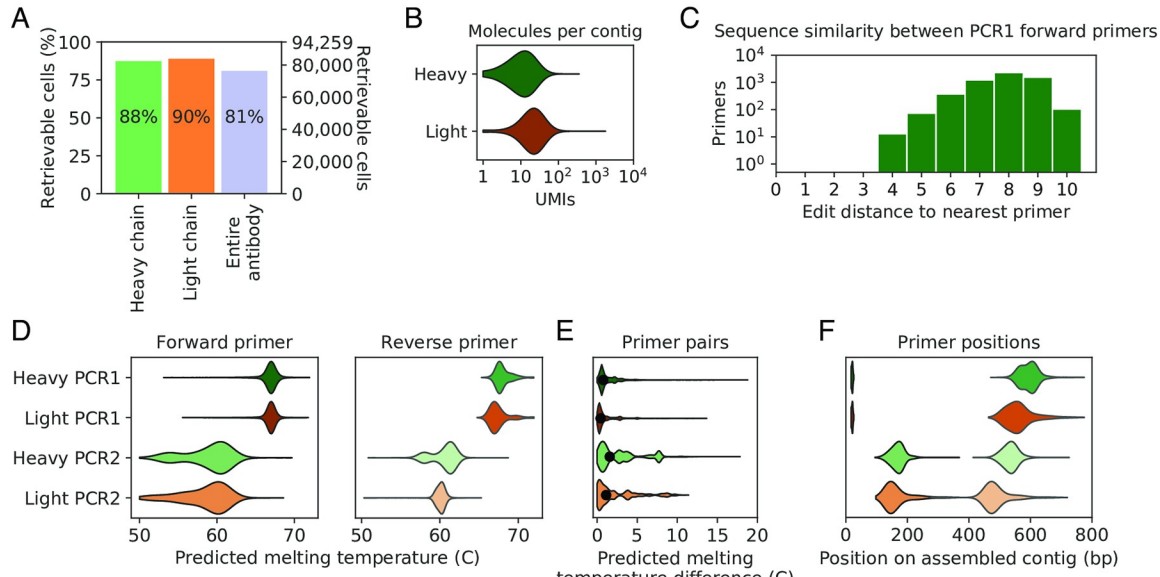

**Fig 2. Computational design and characteristics of SPAR primers.** (A) Success rates of SPAR primer design for heavy chain, light chain, and both chains for all cells in test dataset (n = 94,259 cells). (B) Number of cDNA molecules used to assemble each heavy- and light-chain contig (n = 94,259 heavy-chain contigs; n = 94,259 light-chain contigs). Each molecule carries a distinct unique molecular identifier (UMI). (C) Sequence similarity between PCR1 forward primers for all retrievable heavy-chain genes within one pooled library (n = 5,647 primers), as reflected in the distribution of edit distances to the nearest neighbor. Edit distance is defined as the number of insertions, deletions, or substitutions required to change one sequence to the other, also known as Levenshtein distance. (D) Predicted melting temperature of forward and reverse primers for all retrievable antibodies (n = 76,781). (E) Difference between predicted melting temperatures of forward and reverse primers in each pair for all retrievable antibodies (n = 76,781). Black dot indicates median. (F) Positions of forward and reverse primers for all retrievable antibodies (n = 76,781). Reverse primers are indicated by lighter shading, as in (D).

the nearest sequence is 8 edits away on average, and the most similar pair of PCR1 forward primers is separated by 4 edits. These features suggest that PCR1 primers are sufficiently divergent to support selective amplification.

SPAR primers have favorable properties for PCR. Predicted melting temperatures of PCR1 primers are high (Fig 2D; 67.3 ± 1.1 C, mean ± s.d.) and well matched within pairs (Fig 2E; temperature difference 1.1 ± 1.3 C, mean ± s.d.). PCR2 primer pairs have greater variation in predicted melting temperature due to stronger constraints on primer position (Fig 2D; 59.4 ± 2.4 C, mean ± s.d.), but nevertheless have well matched melting temperature within pairs (Fig 2E; temperature difference 2.5 ± 2.6 C, mean ± s.d.). As designed, PCR2 primers flank the variable region (Fig 2F), permitting one-step cloning into expression vectors. Together, these features suggest that SPAR primers support efficient selective PCR.

## SPAR retrieves full-length antibody variable region cDNA from single cells

To experimentally test retrieval of antibody cDNA, we performed SPAR on 8 target cells chosen at random from the cells with successful primer design within our dataset. We first performed SPAR PCR for these 8 cells. Agarose gel electrophoresis of the SPAR PCR2 products revealed successful retrieval of the expected, full-length antibody heavy- and light-chain variable region cDNA for all 8 targets (100%) (Fig 3B and 3C). We then cloned the cDNA for three of these antibodies into heavy- and light-chain expression vectors using Gibson assembly. Using Sanger sequencing, we confirmed that the target heavy- and light-chain sequences were retrieved for all three cells (Fig 3D). In every case, the retrieved heavy- and light-chain cDNA

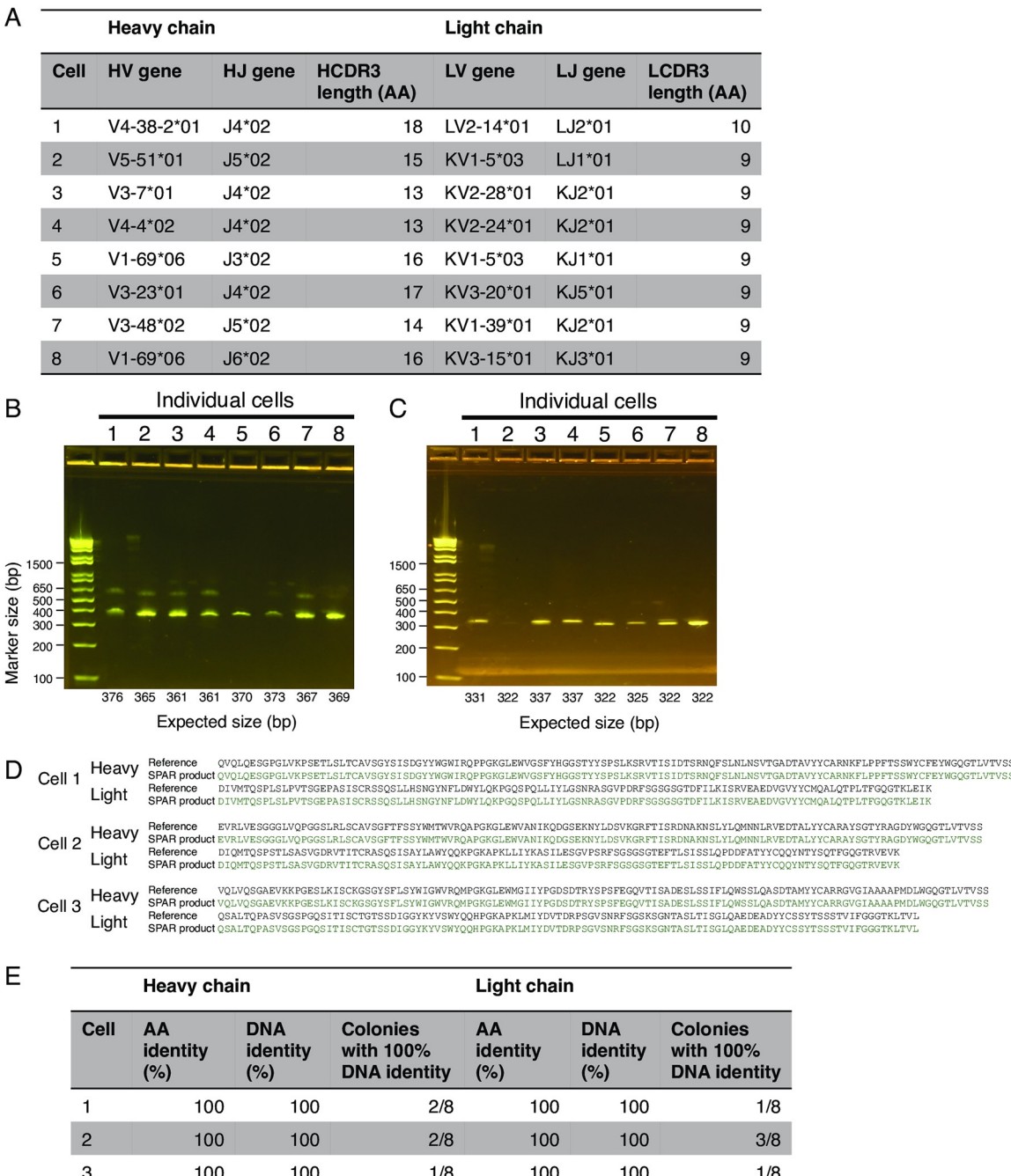

**Fig 3. Retrieval of antibodies from single cells using SPAR.** (A) Molecular features of antibodies targeted for retrieval originating from 8 single cells chosen at random from pooled sequence libraries. Heavy- and light-chain gene usage and CDR3 length are shown. HV, heavy-chain variable; HJ, heavy-chain joining; HCDR3, heavy-chain CDR3; LV, light-chain variable; LJ, light-chain joining; LCDR3, light-chain CDR3; AA, amino acids. (B and C) Agarose gel electrophoresis of SPAR PCR2 products (2% agarose) for heavy- (B) and light-chain genes (C). Expected size of each product is indicated by label at bottom. (D) Sequence verification of SPAR PCR2 products using Sanger sequencing for three cells. Heavy- and light-chain amino acid sequences of SPAR PCR2 products were aligned to the reference sequences obtained by single-cell sequencing. Residues matching the reference are shown in green. (E) Summary of SPAR products for three cells. Colonies were picked after transformation of Gibson Assembly products into *E. coli*, then constructs were Sanger sequenced. For each chain, the fraction of colonies with perfect (100%) identity to reference is shown. One colony with maximal identity to reference was chosen, and the amino acid (AA) and DNA identity of that colony to the reference is shown.

**Table 1. Cost analysis of Selective PCR for Antibody Retrieval (SPAR).**

| Reagent | Supplier | Catalog No. | Cost ($) |
|---|---|---|---|
| Oligo PCR1 Forward primer | IDT | | 5.00 |
| Oligo PCR1 Reverse primer | IDT | | 5.00 |
| Oligo PCR2 Forward primer | IDT | | 10.00 |
| Oligo PCR2 Reverse primer | IDT | | 10.00 |
| KAPA HiFi HotStart ReadyMix | KAPA Biosystems | KK2601 | 3.93 |
| ExoSAP-IT PCR Cleanup Mix | ThermoFisher | 78200 | 0.85 |
| Ampure XP beads | Beckman Coulter | A63880 | 0.19 |
| Total cost per chain | | | 34.97 |
| Total cost per antibody | | | 69.94 |

had perfect amino acid and DNA identity (100% identity) to the target (Fig 3E). These findings demonstrate that SPAR can retrieve full-length antibody cDNA from single cells within pooled libraries of ~5,000 cells with high fidelity and efficiency.

## Discussion

SPAR enables a simple workflow for cloning antibody genes for functional characterization. After surveying antibody sequences at high-throughput using a single-cell sequencing approach, target antibodies can be chosen based on sequence or clonal characteristics, or single-cell phenotypes, such as transcriptome profile [3]. Using SPAR, these antibodies can be cloned and expressed directly from the pooled cDNA library. Our computational analysis suggests that ~81% of human antibodies can be retrieved by SPAR. Notably, many antibodies of practical interest belong to expanded clones, and antibodies that cannot be retrieved by SPAR due to primer constraints may nevertheless be clonally related to antibodies which have highly similar sequence and are retrievable. Furthermore, PCR-based mutagenesis could be used to generate variants in sequence space near retrieved antibodies. SPAR costs ~$70 per antibody (Table 1), which is cheaper than or similar in price to gene synthesis. Importantly, SPAR can be performed within ~29 hours (Table 2), which is faster than the several-week turnaround time of gene synthesis. The speed of SPAR may be advantageous in scenarios requiring rapid response, such as antibody discovery for treatment of emerging infectious disease. Thus, SPAR enables rapid, low-cost cloning and expression of native antibodies from pooled sequencing libraries.

SPAR could be improved and extended in several ways. To improve specificity, the primer design algorithm could explicitly model and penalize potential mispriming within the cDNA pool. To further improve specificity and efficiency, the single-cell sequencing library

**Table 2. Time requirements for Selective PCR for Antibody Retrieval (SPAR).**

| Step | Time (hours) | Hands-on time (hours) |
|---|---|---|
| Design of primers | 1 | 1 |
| Primer synthesis | 24 | |
| SPAR PCR1 | 2 | 0.5 |
| SPAR PCR2 | 2 | 0.5 |
| Total time | 29 | 2 |

Approximate times required to perform each step of the SPAR protocol are indicated. Primer synthesis time typically depends on turnaround time from oligonucleotide synthesis vendors.

preparation procedure could be modified to incorporate a longer sequence barcode. Similar barcoding schemes are used in other single-cell sequencing approaches, such as Drop-seq [8], Microwell-seq [9], and SPLiT-seq [10], and these are also amenable to our approach. For ~10% of heavy- and light-chain genes, PCR primers satisfying standard constraints on predicted melting temperature, G/C content, and secondary structure could not be identified. While we have considered these genes irretrievable, fine-tuning of PCR conditions might allow them to be retrieved. Additionally, usage of longer sequence barcodes during library preparation would increase the likelihood that suitable SPAR PCR1 primers could be designed. Finally, although we successfully retrieved at least one clone with 100% AA and DNA sequence identity to the target in every case during validation (Fig 3E), we noted that other clones had suffered one or a few DNA base substitutions, as revealed by Sanger sequencing. We attribute these errors to the imperfect fidelity of the polymerases used for PCR steps, which could be mitigated by reducing the number of PCR cycles or using higher-fidelity polymerases.

SPAR builds upon previous tag-directed retrieval methods for gene synthesis [11, 12] and enrichment of transcriptomes [13]. The exceptional diversity of natural antibody sequences [14] plays a key role in enabling highly specific nested PCR, permitting retrieval of individual cDNA molecules within complex pools composed of $>10^7$ unique molecules. We anticipate that this approach will facilitate biophysical characterization of antibodies, accelerating antibody discovery and enhancing our understanding of the relationship between antibody sequence and function.

## Materials and methods

### Dataset

For computational analysis and experimental validation, we used our previously published dataset consisting of 94,259 single B cells [3]. Briefly, the subject for this study was a female human aged 18 who was apparently healthy. Subject was vaccinated with the 2011–2012 seasonal trivalent inactivated influenza vaccine, and blood was collected by venipuncture 7 and 9 days afterwards (D7, D9), corresponding to the peak of the memory recall response [15]. Peripheral blood mononuclear cells (PBMCs) were isolated using a Ficoll gradient and frozen according to Stanford Human Immune Monitoring Center protocol.

After thawing, B cells were magnetically enriched using B Cell Isolation Kit II (Miltenyi), then single cells were encapsulated in droplets using 16 lanes of the Chromium device (10X Genomics) with target loading of 14,000 cells per lane. Reverse transcription and cDNA amplification were performed using the Direct Enrichment protocol of the Single Cell 5' V(D)J kit (10X Genomics). All steps were done according to manufacturer's instructions, except with additional cycles of PCR to obtain extra material for protocol testing (19 total cycles). Sequencing libraries were prepared using 50 ng of cDNA as input, then sequenced using the Illumina NextSeq 500 platform with paired-end reads of 150 bp each.

Antibody heavy- and light-chain transcripts were assembled for each cell using cellranger 2.1.0. Single B cells were identified by the presence of a single productive heavy chain and a single productive light chain, yielding a total of 94,259 single B cells.

### Primer design for SPAR

Primer design for SPAR consists of choosing nested PCR primers targeting the antibody gene of interest. In the 10X Genomics VDJ platform, each antibody sequence is typically assembled from sequencing reads from multiple cDNA molecules, which are tagged with the same cell barcode (CBC), but different unique molecular identifiers (UMIs). Accordingly, the gene can be addressed using a primer specific to the CBC and any of the UMIs. To design primers for

PCR1, we first compiled a list of all UMIs supporting the antibody gene assembly, based on the output of cellranger 2.1.0. Full-length cDNA sequences were formed by concatenating partial read 1 primer, CBC, UMI, and template switch oligo (TSO) sequences to the assembled gene (S1 Table). Position of the constant region was determined based on the annotation provided by cellranger 2.1.0. Primers were generated using Primer3 4.1.0 with default parameters except PRIMER_OPT_SIZE = 26; PRIMER_MIN_SIZE = 22; PRIMER_MAX_SIZE = 35; PRIMER_OPT_TM = 67; PRIMER_MIN_TM = 53; PRIMER_MAX_TM = 72; PRIMER_MIN_GC = 30; PRIMER_MAX_GC = 70; PRIMER_SALT_DIVALENT = 2.5; PRIMER_DNA_CONC = 200; PRIMER_PRODUCT_SIZE_RANGE = 250–750. Target was specified as region bounded exclusively by the UMI and constant region, forcing primers to be selected within the CBC and UMI, and the constant region. Primers were allowed to include up to 5 bases of the partial read 1 sequence. Scores of primer pairs were aggregated across all UMIs and the best-scoring primer pair was accepted as the PCR1 primers.

We then designed primers for PCR2 that flank the antibody variable region. We determined the amplicon sequence produced by PCR1, then located the variable region sequence using IgBlast [16]. Primers were generated using Primer3 4.1.0 with default parameters except PRIMER_OPT_SIZE = 20; PRIMER_MIN_SIZE = 13; PRIMER_MAX_SIZE = 35; PRIMER_MIN_TM = 50; PRIMER_OPT_TM = 60; PRIMER_MAX_TM = 70; PRIMER_MIN_GC = 30; PRIMER_MAX_GC = 70; PRIMER_SALT_DIVALENT = 2.5; PRIMER_DNA_CONC = 200; PRIMER_PRODUCT_SIZE_RANGE = 100–700. The best-scoring primer pair was accepted as the PCR2 primers.

This workflow was implemented using custom Python scripts. For each individual cell, the workflow was carried out separately for the heavy- and light-chain genes.

## Computational analysis of retrievability of human antibody repertoire

To assess how much of the human antibody repertoire can be retrieved, we performed SPAR primer design for all single cells in our dataset (n = 94,259). We used the default parameters for SPAR primer design stated in the previous section. An antibody was defined as retrievable if acceptable primers were found for PCR1 and PCR2 for both heavy- and light-chain genes. To assess the sequence similarity between PCR1 forward primers, we calculated the edit distance, also known as the Levenshtein distance, between all pairs of acceptable PCR1 forward primers for one pooled library (chosen at random). Predicted melting temperatures were calculated using Primer3. Data visualization and analysis were performed using JupyterLab [17].

## Experimental validation of SPAR

To demonstrate that SPAR enables retrieval of single-cell antibody cDNA from pooled libraries, we chose 8 cells at random from our dataset. For each cell, SPAR primers designed using the above workflow were synthesized (IDT).

PCR1 was performed using 12.5 uL of HiFi ReadyMix 2X (Kapa Biosystems), 0.75 uL each of forward and reverse primer (final concentration 0.3 uM each), 1 uL of template, and 10 uL of water. Template was 0.5 ng of cDNA from single-cell sequencing library preparation. PCR1 protocol was 95 C for 3 min; 15 cycles of 98 C for 20 sec, 65 C for 15 sec, 72 C for 1 min; 72 C for 1 min. Primers from PCR1 were then degraded by adding 5 uL of PCR1 product to 2 uL of ExoSAP-IT (ThermoFisher), and incubating at 37 C for 15 min, then 80 C for 15 min.

PCR2 was performed using the same conditions, except using 1 uL of previous product as template. PCR2 protocol was 95 C for 3 min; 15 cycles of 98 C for 20 sec, 51 C for 15 sec, 72 C for 1 min; 72 C for 1 min. Products were visualized by electrophoresis using E-Gel EX 2% agarose gels (ThermoFisher).

To demonstrate one-step cloning of SPAR products into expression vectors and verify their sequences, we cloned the PCR2 products using Gibson assembly and performed Sanger sequencing. Heavy- and kappa-chain expression vectors were VRC01 CMV/R HC and VRC01 CMV/R LC [18], respectively. Lambda-chain expression vector was VRC01 CMV/R Lambda LC, which was created by replacing the kappa constant region with the lambda constant region in VRC01 CMV/R LC [18]. The reagent was obtained through the NIH AIDS Reagent Program, Division of AIDS, NIAID, NIH: CMVR VRC01 H/L, from Dr. John Mascola. Vectors were linearized by PCR. PCR conditions were the same as above, except 35 cycles were performed with annealing at 70 C, extension for 6 min, and final extension for 6 min. Template was 1 ng of vector. Products were purified using Ampure XP beads (Agencourt) at 0.7 bead to product volume ratio. Gibson assembly was performed using 10 uL of Gibson Assembly Master Mix (NEB), ~150 fmol insert, and ~50 fmol vector in a total volume of 20 uL. Products were transformed into E. cloni 10G Supreme cells (Lucigen) by electroporation following manufacturer's instruction. After overnight growth, eight colonies were picked and cultured in LB with 50 ug/mL kanamycin for 2 hours. Templates for colony PCR were prepared by diluting 10 uL of culture in 90 uL of water, then incubating at 95 C for 10 min. Colony PCR was performed using 12.5 uL of HiFi ReadyMix 2X (Kapa Biosystems), 0.75 uL each of forward and reverse sequencing primers (SeqF and SeqHR, SeqLR, or SeqKR; final concentration 0.3 uM each), 1 uL of template, and 10 uL of water. PCR protocol was 95 C for 3 min; 35 cycles of 98 C for 20 sec, 55 C for 15 sec, 72 C for 1 min; 72 C for 1 m. Sanger sequencing was performed on products using sequencing primers (Molecular Cloning Laboratories).

## Supporting information

**S1 Table. Oligonucleotide sequences used in this study.**
(XLSX)

**S1 Raw images.**
(PDF)

## Acknowledgments

We thank Peter Kim, Payton Weidenbacher, and Natalia Friedland for providing heavy- and light-chain expression vectors.

## Author Contributions

**Conceptualization:** Felix Horns.

**Funding acquisition:** Stephen R. Quake.

**Investigation:** Felix Horns.

**Methodology:** Felix Horns.

**Software:** Felix Horns.

**Supervision:** Stephen R. Quake.

**Validation:** Felix Horns.

**Visualization:** Felix Horns.

**Writing – original draft:** Felix Horns.

**Writing – review & editing:** Felix Horns.

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
