## [Decision Letter · Decision Letter 0]

11 Jun 2020

PONE-D-20-13444

Cloning antibodies from single cells in pooled sequence libraries by selective PCR

PLOS ONE

Dear Dr. Horns,

Thank you for submitting your manuscript to PLOS ONE. After careful consideration, we feel that it has merit but does not fully meet PLOS ONE’s publication criteria as it currently stands. Therefore, we invite you to submit a revised version of the manuscript that addresses the points raised during the review process.

We look forward to receiving your revised manuscript.

Kind regards,

Kevin A. Henry

Academic Editor

PLOS ONE

Journal Requirements:

3. We note that you have a patent relating to material pertinent to this article. Please provide an amended statement of Competing Interests to declare this patent (with details including name and number), along with any other relevant declarations relating to employment, consultancy, patents, products in development or modified products etc. Please confirm that this does not alter your adherence to all PLOS ONE policies on sharing data and materials, as detailed online in our guide for authors http://journals.plos.org/plosone/s/competing-interests by including the following statement: "This does not alter our adherence to  PLOS ONE policies on sharing data and materials.” If there are restrictions on sharing of data and/or materials, please state these. Please note that we cannot proceed with consideration of your article until this information has been declared.

Additional Editor Comments (if provided):

Reviewers' comments:

Reviewer's Responses to Questions

**Comments to the Author**

1. Is the manuscript technically sound, and do the data support the conclusions?

Reviewer #1: Yes

Reviewer #2: Yes

Reviewer #3: Yes

2. Has the statistical analysis been performed appropriately and rigorously? 

Reviewer #1: Yes

Reviewer #2: N/A

Reviewer #3: Yes

3. Have the authors made all data underlying the findings in their manuscript fully available?

Reviewer #1: Yes

Reviewer #2: Yes

Reviewer #3: Yes

4. Is the manuscript presented in an intelligible fashion and written in standard English?

Reviewer #1: Yes

Reviewer #2: Yes

Reviewer #3: Yes

5. Review Comments to the Author

Reviewer #1: The manuscript by Horns and Quake presents an elegant method to retrieve monoclonal antibody sequences (VH and VL) from cDNA high-throughput sequencing data for subsequent production of recombinant antibodies. The method relies on the use of single-cell barcodes when performing single-cell sequencing to specifically amplify the antibody sequences of interest, by nested PCR. The authors first describe the computational methodology then moved onto a functional example from a 10x Genomics 5' V(D)J sequencing experiment.

The manuscript is clearly written and will be helpful for a number of scientists performing single-cell antibody NGS using the 10X Genomics 5' V(D)J kit and who would like to further characterise some antibodies sequenced.

Minor comments:

(1) Did the paired heavy-light chain sequencing cover the whole VH and VL sequences or just some CDRs? If only some CDRs are being sequenced, would this mean that some framework (FR) regions be missed? In particular the FR1, where the primers for PCR2 will bind.

(2) Why can SPAR primers be designed for only 88% of VH and 90% of VL? Can the barcodes misprime to other cDNA sequences? Or is it due to poor sequencing of the framework 1 or the VH or VL?

(3) Why was the retrieval and cloning of the VH and VL so inefficient in the end? Figure 3E shows that between 1 and 3 out of 8 clones had 100% DNA sequence identity

Reviewer #2: Summary: The authors describe an approach to recover specific antibody amplicons from a pool of bulk cDNA for cloning and expression. Overall the manuscript is clearly written and easy to follow. The introduction, however, emphasizes that functional evaluation of antibodies is a bottleneck, which is true, but the proposed approach only addresses one piece of the functional screen bottleneck, which is cloning. It's a bit overstated to imply that the proposed approach addresses the functional screen bottleneck. Additionally, while I appreciate the arguments being made, I find tables 1 and 2 as not adding major contributions to the paper. Table 1 for example doesn't include personnel time. The cost of ordering the heavy and light chain genes for synthesis is ~$100, so the cost argument isn't super strong given how easy it is to order genes these days. Second, Table 2 is a bit subjective - would 1 hr of hands on time for primer design be consistent for everyone? In many ways, it is much easier to just order the gene for synthesis. I think there's still value to the approach, however, for anyone who really wants to do an immediate PCR to recover HC+LC genes instead of waiting for gene synthesis, but I don't think this needs to be overhyped. I still think worth publishing in plos one.

Reviewer #3: The authors report a very clever technique for rapidly extracting individual mAbs of interest from bulk 10x Genomics sequencing libraries. The manuscript is well written, the analyses and experiments are technically sound, and the data is clearly presented. I recommend acceptance and publication without any modification.

6. PLOS authors have the option to publish the peer review history of their article (what does this mean?). If published, this will include your full peer review and any attached files.

Reviewer #1: No

Reviewer #2: No

Reviewer #3: Yes: Bryan Briney

---

## [Author Response · Author response to Decision Letter 0]

25 Jun 2020

Response to editor comments

We have revised the manuscript to conform to the style requirements.

All original gel images are now included as S1_raw_images. These images were used to generate Figure 3B and Figure 3C. We are providing this image data as Supplementary Information. We confirm that the gels reported in the main figures were prepared according to the guidelines. Specifically, no manipulation of the images was performed except for cropping empty lanes.

3. We note that you have a patent relating to material pertinent to this article. Please provide an amended statement of Competing Interests to declare this patent (with details including name and number), along with any other relevant declarations relating to employment, consultancy, patents, products in development or modified products etc. Please confirm that this does not alter your adherence to all PLOS ONE policies on sharing data and materials, as detailed online in our guide for authors http://journals.plos.org/plosone/s/competing-interests by including the following statement: "This does not alter our adherence to PLOS ONE policies on sharing data and materials.” If there are restrictions on sharing of data and/or materials, please state these. Please note that we cannot proceed with consideration of your article until this information has been declared.

We are preparing to submit a patent application related to this manuscript. We will provide an amended statement of Competing Interests as soon as the application is submitted and the patent name and number are available. This does not alter our adherence to PLOS ONE policies on sharing data and materials, and there are no restrictions on sharing of data or materials. We confirm that all authors do not have potential competing interests.

4. PLOS requires an ORCID iD for the corresponding author in Editorial Manager on papers submitted after December 6th, 2016. Please ensure that you have an ORCID iD and that it is validated in Editorial Manager.

We are happy to add the ORCID iDs for all authors. However, when attempting to link the ORCID iD in Editorial Manager, we received a notice “The ORCID site reported the following error and the authorization process cannot be completed:” (The rest of the error message was blank). The ORCID iD of corresponding author Stephen Quake is https://orcid.org/0000-0002-1613-0809, and the ORCID iD of author Felix Horns is https://orcid.org/0000-0001-5872-5061.

Response to reviewers

Referee 1

(1) Did the paired heavy-light chain sequencing cover the whole VH and VL sequences or just some CDRs? If only some CDRs are being sequenced, would this mean that some framework (FR) regions be missed? In particular the FR1, where the primers for PCR2 will bind.

The paired heavy- and light-chain sequencing covers the entire VH and VL sequences. Thus, no framework regions are missed. We have added a sentence explaining this (Line 100). 

(2) Why can SPAR primers be designed for only 88% of VH and 90% of VL? Can the barcodes misprime to other cDNA sequences? Or is it due to poor sequencing of the framework 1 or the VH or VL?

For ~10% of heavy- and light-chain genes, SPAR PCR primers satisfying constraints on predicted melting temperature, G/C content, and secondary structure could not be found. While we have considered these genes irretrievable, fine-tuning of PCR conditions might allow them to be retrieved. Additionally, if we used a library preparation method that incorporated a larger CBC and UMI, then there would be a higher likelihood that suitable SPAR PCR1 primers could be designed. We have added a comment on these sources of irretrievability and strategies to mitigate them in the Discussion (Line 183).

(3) Why was the retrieval and cloning of the VH and VL so inefficient in the end? Figure 3E shows that between 1 and 3 out of 8 clones had 100% DNA sequence identity

Nearly all of the clones without perfect DNA sequence identity to the target had suffered one or a few single base changes. We attribute this to the imperfect fidelity of the polymerases used for the various PCR steps. This issue could be mitigated by reducing the number of cycles of PCR or using higher fidelity polymerases. We have added a comment about this to the Discussion (Line 189).

Referee 2

The introduction, however, emphasizes that functional evaluation of antibodies is a bottleneck, which is true, but the proposed approach only addresses one piece of the functional screen bottleneck, which is cloning. It's a bit overstated to imply that the proposed approach addresses the functional screen bottleneck. 

We agree that the functional evaluation of antibodies faces further bottlenecks downstream of cloning and expression. We have amended the main text in several places to emphasize that our method addresses the cloning bottleneck (Line 70 in the introduction and Line 162 in the discussion).

Additionally, while I appreciate the arguments being made, I find tables 1 and 2 as not adding major contributions to the paper. Table 1 for example doesn't include personnel time. The cost of ordering the heavy and light chain genes for synthesis is ~$100, so the cost argument isn't super strong given how easy it is to order genes these days. 

We have clarified that the cost of SPAR is cheaper than or similar in price to gene synthesis (Line 174).

Second, Table 2 is a bit subjective - would 1 hr of hands on time for primer design be consistent for everyone? In many ways, it is much easier to just order the gene for synthesis. I think there's still value to the approach, however, for anyone who really wants to do an immediate PCR to recover HC+LC genes instead of waiting for gene synthesis, but I don't think this needs to be overhyped. 

We agree that the speed of SPAR is a valuable aspect. We have edited the text to emphasize that the turnaround time is faster than gene synthesis and to highlight the situations where this might be advantageous (Line 176). With respect to Table 2, while it is true that the times are subjective, we believe that it is helpful for practitioners of the method to understand the approximate timelines. This is often presented in molecular biology methods papers. We also believe that these timelines, while approximate, illustrate that the timescale is substantially faster than gene synthesis. Nevertheless, we have now indicated in the Table legend that these times are approximate.

---

## [Decision Letter · Decision Letter 1]

9 Jul 2020

Cloning antibodies from single cells in pooled sequence libraries by selective PCR

PONE-D-20-13444R1

Dear Dr. Horns,

We’re pleased to inform you that your manuscript has been judged scientifically suitable for publication and will be formally accepted for publication once it meets all outstanding technical requirements.

Kind regards,

Kevin A. Henry

Academic Editor

PLOS ONE

Additional Editor Comments (optional):

I am pleased to say that the reviewers have found the revised manuscript suitable for publication in PLoS One. Congratulations on the very nice work. Reviewer 2 contacted me by email to confirm that their concerns had been addressed.

Reviewers' comments:

Reviewer's Responses to Questions

**Comments to the Author**

1. If the authors have adequately addressed your comments raised in a previous round of review and you feel that this manuscript is now acceptable for publication, you may indicate that here to bypass the “Comments to the Author” section, enter your conflict of interest statement in the “Confidential to Editor” section, and submit your "Accept" recommendation.

Reviewer #1: All comments have been addressed

2. Is the manuscript technically sound, and do the data support the conclusions?

Reviewer #1: Yes

3. Has the statistical analysis been performed appropriately and rigorously? 

Reviewer #1: Yes

4. Have the authors made all data underlying the findings in their manuscript fully available?

Reviewer #1: Yes

5. Is the manuscript presented in an intelligible fashion and written in standard English?

Reviewer #1: Yes

6. Review Comments to the Author

Reviewer #1: All the comments have been properly addressed. The manuscript should be accepted as is, without any further improvements.

7. PLOS authors have the option to publish the peer review history of their article (what does this mean?). If published, this will include your full peer review and any attached files.

Reviewer #1: No

---

## [Editor Report · Acceptance letter]

17 Jul 2020

PONE-D-20-13444R1 

Cloning antibodies from single cells in pooled sequence libraries by selective PCR 

Dear Dr. Horns:

I'm pleased to inform you that your manuscript has been deemed suitable for publication in PLOS ONE. Congratulations! Your manuscript is now with our production department. 

Kind regards, 

on behalf of

Dr. Kevin A. Henry 

Academic Editor

PLOS ONE